# Effect of Microalgae and Macroalgae Extracts on Non-Alcoholic Fatty Liver Disease

**DOI:** 10.3390/nu13062017

**Published:** 2021-06-11

**Authors:** Maitane González-Arceo, Saioa Gómez-Zorita, Leixuri Aguirre, María P. Portillo

**Affiliations:** 1Nutrition and Obesity Group, Department of Pharmacy and Food Science, Faculty of Pharmacy and Lucio Lascaray Research Center, University of the Basque Country (UPV/EHU), 01008 Vitoria-Gasteiz, Spain; maitane.gonzalez@ehu.eus (M.G.-A.); mariapuy.portillo@ehu.eus (M.P.P.); 2Bioaraba Health Research Institute, 01006 Vitoria-Gasteiz, Spain; 3CIBER Fisiopatología de la Obesidad y Nutrición (CIBERobn), Instituto de Salud Carlos III (ISCIII), 28222 Madrid, Spain

**Keywords:** non-alcoholic fatty liver disease, liver steatosis, macroalgae, microalgae

## Abstract

The present review aims to gather scientific evidence regarding the beneficial effects of microalgae and macroalgae extracts on non-alcoholic fatty liver disease (NAFLD). The described data show that both microalgae and macroalgae improved this alteration. The majority of the reported studies analysed the preventive effects because algae were administered to animals concurrent with the diet that induced NAFLD. The positive effects were demonstrated using a wide range of doses, from 7.5 to 300 mg/kg body weight/day or from 1 to 10% in the diet, and experimental periods ranged from 3 to 16 weeks. Two important limitations on the scientific knowledge available to date are that very few studies have researched the mechanisms of action underlying the preventive effects of microalgae on NAFLD and that, for the majority of the algae studied, a single paper has been reported. For these reasons, it is not possible to establish the best conditions in order to know the beneficial effects that these algae could bring. In this scenario, further studies are needed. Moreover, the beneficial effects of algae observed in rodent need to be confirmed in humans before we can start considering these products as new tools in the fight against fatty liver disease.

## 1. Introduction

Non-alcoholic fatty liver disease (NAFLD) covers a wide spectrum of histopathological abnormalities ranging from simple steatosis to steatohepatitis (NASH). Hepatic steatosis is the most benign and common form of NAFLD that is defined as intrahepatic fat accumulation of at least 5% of liver weight. However, this condition can evolve to more advanced stages if hepatocytes are exposed to stress, causing cell death, apoptosis, inflammation and fibrosis and leading to NASH. This NASH can result in cirrhosis and hepatocellular carcinoma [1,2]. The prevalence of NAFLD shows a high variability, ranging from 6 to 35% in the general population. These rates are experiencing an upward trend due to the current epidemic of obesity [3] and type 2 diabetes [2]; in fact, about 50% of NAFLD patients and 80% of patients with NASH are obese [4].

NAFLD is supposed to occur in patients with no alcohol or little alcohol consumption. Some authors have suggested that the term NAFLD overemphasizes the “non-alcoholic” aspect, and due to its association with a greater number of co-morbidities, they have proposed substituting NAFLD with metabolic associated fatty liver disease (MAFLD) [5,6].

Currently, there is no specific treatment for liver steatosis. The first step in its management consists of lifestyle intervention with caloric intake restriction and exercise. However, patients find it difficult to implement and achieve these lifestyle modifications. At present, no drugs have yet been approved, and pharmacological treatment devotes efforts to associated co-morbidities that contribute to the pathogenesis of NAFLD, such as, obesity, type 2 diabetes mellitus or dyslipidemia [4]. Due to the increasing prevalence and treatment limitations of NAFLD, there is an urgent need to seek new sources of bioactive compounds with potential preventive and/or therapeutic action.

In this context, there is increasing research on algae, based on the fact that many cultures have traditionally used them as food and medicine, and that 70% of the earth’s surface is covered by the ocean, which may provide additional opportunities in view of the limitation of terrestrial resources. Algae are typical of Japanese, Korean and Chinese dietary patterns, and more recently, seaweeds and seaweed-based foodstuffs, such as sushi, are commonly found in western societies. Moreover, they are an important component of the Atlantic dietary pattern. Significantly, algae represent a good source of proteins, vitamins, minerals and fiber [7,8,9,10]. In addition, they are rich in a great number of bioactive compounds, such as peptides, pigments, phenolic compounds and fatty acids with potential applications in health, due to their antioxidant, antimicrobial, anti-inflammatory, anticancer, antidiabetic, antihypertensive, antihiperlipidaemic and antiobesity effect [11,12]. Algae can also be used as ingredients to be included, dehydrated or powdered, in functional foods. Indeed, they can have a relevant role in reformulation strategies for the manufacture of technologically and sensorily viable food stuffs with health-promoting compositions due to their technological, organoleptic, and nutritional properties. An example is the reformulation of meat products (frankfurters, patties and restructured steaks) by adding nori, wakame or sea spaghetti seaweeds, reported by Cofrades et al. [13].

To the best of our knowledge, no in vitro studies with algae extracts in cultured hepatocytes have been carried out to date. The present review aims to gather scientific evidence regarding the beneficial effect of microalgae and macroalgae extracts on hepatic steatosis in preclinical and clinical studies, as well as the potential mechanisms involved in these effects.

## 2. Animal Studies

The vast majority of algae extract effects have been studied in rodent models. Figure 1 summarizes the main effects observed in the present review.

### 2.1. Microalgae

As far as we know, only four studies analysing the effects of microalgae extracts on NAFLD in animal models have been published to date. (Table 1).

Kumar et al. [14] studied the effect of blending two microalgae, *Scenedesmus dimorphus* and *Schroederiella apiculata*, which contained 46.1% protein, 19.6% insoluble fibre and 2.8% omega-3 fatty acids. For this purpose, Wistar rats were distributed into four experimental groups: the control group was fed with a corn starch diet (C) containing 68% carbohydrates as polysaccharides; a second group received a high-carbohydrate high-fat diet (H) where fat, fructose and sucrose represented 24%, and drinking water contained 25% fructose (total 68% carbohydrates in food and water); and the remaining two groups were fed with the corn starch diet supplemented with 5% *Scenedesmus dimorphus* and *Schroederiella apiculata* blend (CSC), or the high-carbohydrate high-fat diet supplemented with 5% *Scenedesmus dimorphus* and *Schroederiella apiculata* mixture (HSC). Although the total experimental period length was 16 weeks, the microalgae combination was only provided during the last eight weeks. As expected, and compared to the C group, the rats fed with the high-carbohydrate high-fat diet showed increased liver weight, higher presence of enlarged fat vacuoles and higher infiltration of inflammatory cells. Interestingly, these effects were totally prevented by the inclusion of the microalgae mixture into the diet. In line with these effects, and compared to the control group, the increased values observed in alanine aminotransferase (ALT) and aspartate aminotransferase (AST) in the H group, which indicated liver damage, were normalized. Moreover, improved glucose tolerance and insulin sensitivity, which are closely related to steatosis development, were observed in the groups supplemented with the microalgae mixture. No effects induced by the microalgae extract were detected in the CSC group because, due to the standard diet composition provided to these rats, all liver parameters were normal. 

Nakashima et al. [15] studied the effects of *Euglena gracilis* (29.4% carbohydrate, 42.3% protein and 19% fat) using STAM (Stelic Animal Model) mice, a model of non-alcoholic steatohepatitis, induced by streptozocin. Mice were fed with a high-fat diet and *Euglena gracilis* was orally administered, at a dose of 3 g/kg body weight, for 27 days. At the end of the experimental period, liver weight, liver triglycerides and plasma ALT levels were measured, and no differences were observed between the *Euglena gracilis*-treated group and the control group. Rats treated with *Euglena gracilis* showed lower hepatic fibrosis than the controls. Although fibrosis is one of the parameters involved in the calculation of the NAFLD activity score (NAS score), this index did not differ among rats receiving the microalga and the vehicle. The gene expression of the inflammation-related genes interleukin 1 *β* (*Il-1β*)*,* interleukin 6 (*Il-6*), tumour necrosis factor α (*Tnf-α*) and monocyte chemoattractant protein 1 (*Mcp-1*), as well as fibrogenic markers such as *α-Sma,* collagen type I alpha 2 chain (*Col1a2*) and collagen type III alpha 1 chain (*Col3a1*) remained unchanged. The authors concluded that *Euglena gracilis* may not be involved in inflammation, although it may be effective in inhibiting the activation of hepatic stellate cells, thus attenuating collagen overproduction. 

Pham et al. [16] studied the antifibrotic effect of the blue-green microalga *Spirulina platensis* in a mouse model of diet-induced fibrosis. C57BL/6J mice were assigned to three groups: a group fed with a control low-fat diet (LF; 6% fat), a second fed with a high-fat/high-sucrose/high-cholesterol diet (HF; 34%/38%/2%), and a third group fed with the HF diet supplemented with 2.5% *Spirulina platensis* (HF/SP) for 20 weeks. Liver weight was significantly higher in the HF and HF/SP groups than in the LF group. Although *Spirulina platensis* supplementation did not modify liver triglyceride content, it reduced plasma ALT level at 16 weeks. Mice in HF and HF/SP groups displayed collagen accumulation in hepatocytes, data which are in accordance with mRNA levels of collagen type I alpha 1 chain (*Col1a1*). Regarding serum parameters, although *Spirulina platensis* supplementation did not prevent the increment produced by the HF diet in glucose concentrations, it improved slightly glucose tolerance, which is a positive effect taking into account that a bad glycaemic control is a risk factor for liver steatosis development. 

The spleen is a secondary lymphoid organ, closely associated with the liver, via the portal vein system. It is a source of inflammatory cells that migrate to the liver upon liver injury, thus contributing to the development of liver fibrosis. Spleen weight was increased in the HF group, and *Spirulina platensis* supplementation did not prevent this effect. When splenocytes were isolated, mice receiving *Spirulina platensis* supplementation had significantly lower gene expression level of basal *Il-1β* and a trend to decreased expression of *Il-6*. When splenocytes were tested ex vivo for their lipopolysaccharide (LPS) sensitivity, the results showed that the cells from mice fed with the supplemented diet had significantly lower mRNA levels of *Il-1β* and *Tnf-α*, after LPS induction, while *Il-6* expression level remained unchanged. The authors suggested that *Spirulina platensis,* at the dose used in their study, exerted anti-inflammatory effects, although it did not prevent the fibrosis development induced by the HF diet.

Mayer et al. [17] have recently studied the preventive effects of *Tisochrysis lutea* (Tiso) on metabolic alterations associated with obesity, including NAFLD. Wistar rats were distributed into three experimental groups and were fed with a standard diet (CTRL), a high-fat high-fructose diet (HF) with 10% fructose in drinking water, or the HF diet but supplemented with 12% of *Tisochrysis lutea* (HF-Tiso), for eight weeks. At the end of the experimental period, and compared to other groups, the HF-Tiso group showed decreased values of body weight, abdominal and epididymal adipose tissues, liver triglyceride content, and plasma AST level, but no changes in ALT level were identified. Accordingly, the HF-Tiso group showed a lower AST/ALT ratio. The alga supplementation also lowered plasma glucose, insulin and the homeostatic model assessment for insulin resistance (HOMA-IR) index, meaning that insulin resistance was reduced. The pro-inflammatory cytokines TNF-α and IL-6 were increased in the HF group and *Tisochrysis lutea* supplementation significantly prevented the effect on TNF-α, without affecting IL-6.

To sum up, after reviewing the literature concerning the effects of microalgae on NAFLD, it can be pointed out that although the results reported are encouraging, the number of studies is still very scarce. In addition, the reported studies address the effects of different microalgae, and thus only one source of information is available for each one. According to the experimental design used, all the reported studies analysed the preventive effects of microalgae because these were included in the diet that induced liver alterations. Although in all of them microalgae improved several alterations, there is a lack of consensus on the specific effects observed. Thus, whereas the three studies that analysed this parameter reported a reduction of liver inflammation, only two of these studies described a reduction of liver triglyceride content (the other two studies did not find significant changes). Moreover, two of the studies analysed the effects of fibrosis and only one observed a significant improvement. In this scenario, further research is needed in order to assess the effects of microalgae on NAFLD.

### 2.2. Macroalgae

Several studies have analysed the effects of green, red and brown macroalgae on NAFLD using different animal models.

#### 2.2.1. Green Algae

Two green algae have been studied in the reported literature: *Caulerpa lentillifera* and *Ulva prolifera* (Table 2).

The effect of the green seaweed *Caulerpa lentillifera* in C57BL/6J mice was studied by Sharma et al. [18]. Mice were distributed into three experimental groups: control group (fed with a standard diet), HFD group (fed with a diet containing 60% of energy as fat) and HFD + CL fed with the same high-fat diet and 250 mg/kg body weight of the algae, administered by oral gavage, for ten weeks. HFD increased plasma levels of free fatty acids, glucose and insulin, and the addition of the algae extract to the diet led to decreased levels of these parameters. HFD produced increases in liver weight and hepatic triglycerides. The administration of *Caulerpa lentillifera* prevented all these effects.

Du Preez et al. [19] studied the effects of the same seaweed in Wistar rats. Animals were distributed in four groups: two groups received either corn starch (C) or high-carbohydrate high-fat (H) diets for 16 weeks. The other two groups received C or H diets for the first eight weeks and after, they were fed on the same diets but supplemented with 5% dried *Caulerpa lentillifera* (44% carbohydrates, 14% lipids, 7% protein and 17.5% fibre) for the remaining eight weeks (CCL and HCL groups, respectively). Compared to the C group, liver fat deposition was higher among rats from the H group, and the addition of *Caulerpa lentillifera* to the diet partially prevented this effect. However, infiltration of inflammatory cells and plasma activities of ALT and AST did not differ among the experimental groups. The analysis of microbiota composition revealed that there were no differences in diversity and richness among the four experimental groups. However, rats fed with the diets supplemented with *Caulerpa lentillifera* showed lower *Firmicutes/Bacteroidetes* ratios than rats fed with a high-carbohydrate high-fat diet. The authors concluded that these changes in gut microbiota could be a mechanism that justified the improvement of the above-mentioned parameters in the HCL group, as they found a strong correlation between bacterial community structure and oral glucose tolerance test, liver weight, retroperitoneal, epididymal and total abdominal fat.

Song et al. [20] conducted a study using C57BL/6J mice. The control group was fed with a standard diet, another group was administered a high-fat diet (60% fat; HFD group) and the remaining two groups received the same diet but with 2 or 5% of an *Ulva prolifera* ethanol extract added to the drinking water (HFD2 and HFD5 groups, respectively). After eight weeks, hepatic triglyceride content, as well as serum insulin was increased following high-fat feeding. All these deleterious effects were prevented by the administration of *Ulva prolifera* (at both doses). Similarly, the impairment of glucose tolerance and the reduction in insulin sensitivity were also prevented. In order to determine some of the mechanisms of action that explain the anti-steatosis effect of *Ulva prolifera*, the authors analysed liver gene expression of diacylglycerol O-acyltransferase 1 (*Dgat1*) and diacylglycerol O-acyltransferase 2 (*Dgat2*), two enzymes involved in triglyceride assembly, and in line with the results concerning liver triglycerides, the increase induced by the high-fat diet in these parameters was prevented by both doses of the seaweed. Moreover, gene expression of enzymes involved in fatty acid oxidation, a process that consumes this lipid species avoiding its availability for triglyceride synthesis, were also analysed. In this case, and compared to the control group, mRNA levels of carnitine palmitoyltransferase 1A (*Cpt-1a*)*,* medium-chain acyl-CoA dehydrogenase (*Acadm*) and acyl-CoA oxidase 1 (*Acox1*) were decreased in the HFD group, and yet again, this effect was prevented by both doses of *Ulva prolifera*. Concerning inflammation-related parameters, serum IL-1β, IL-6 and TNF-α concentrations, and liver gene expression of *Il-1β, Il-6* and *Tnf-α* were determined. Higher values were found in the HFD group, and this effect was prevented in both the HFD2 and the HFD5 groups. With regard to oxidative stress, the negative effect induced by the high-fat diet on reactive oxygen species (ROS) content, glutathione (GSH) content and glutathione peroxidase (GSHPx) activity were arrested by both doses of *Ulva prolifera*. A dose-response pattern of reaction was not found in these effects.

In summary, as in the case of microalgae, the number of studies addressing the preventive effect of green algae on NAFLD is still scarce. The three reported works (two of them carried out with *Caulerpa lentillifera*) revealed a reduction in hepatic triglyceride accumulation when animals were fed with a high-fat diet, but only one of the studies addressed several aspects of the potential mechanisms of action (insulin resistance, fatty acid and triglyceride metabolism, oxidative stress). In one of these pieces of research, the authors analysed the effects of seaweed supplementation on gut microbiota composition, but they did not establish a clear relationship between these changes and the effects on liver steatosis, just a significant correlation.

#### 2.2.2. Red Algae

Five red algae were studied in the reported literature: Plocamium telfairiae, Palmaria mollis, Sarconema filiforme Grateloupia elliptica and Gromphadorhina oblongata (Table 3).

Using C57BL/6 mice, Kang et al. [21] analysed the effects of *Plocamium telfairiae*, administered for 14 weeks. Animals were distributed into three groups: the control group was fed with a standard diet, the HFD group received a high-fat diet and the HFD + PL group was administered the same high-fat diet but supplemented with 100 mg/kg body weight of *Plocamium telfairiae*. Hepatic steatosis was induced in the HFD group, and compared to the controls, this deleterious effect was partially prevented by *Plocamium telfairiae*. In serum, glucose was also increased in the HFD group, and the seaweed supplementation totally prevented these effects.

Using the same algae and the same mice model, in the study of Lu et al. [22] mice were distributed into five groups and were fed with experimental diets for seven weeks: a control group was fed with a chow diet, a high-fat diet group (HFD) received a high-fat diet (45% fat), a PTE100 group was administered the high-fat diet but supplemented with 100 mg/kg body weight/day of *Plocamium telfairiae*, a PTE165 group was fed with the high-fat diet but supplemented with 165 mg/kg body weight/day of *Plocamium telfairiae* and a PTE300 group was administered the high-fat diet but supplemented with 300 mg/kg body weight/day of *Plocamium telfairiae*. Compared to the control group, hepatic histological analysis revealed increased hepatic steatosis in HFD mice, which in turn was reduced in mice fed with the algae. The authors did not indicate if a dose-response pattern was observed.

Nakayama et al. [23] studied the potential effects of the red algae *Palmaria mollis* using NSY/HOS mice, assigned to three experimental groups for four weeks. Mice were fed with a standard diet or a high-fat diet, supplemented or not with the red alga *Palmaria mollis* (2.5% w/w). At the end of the study, and compared to the mice fed with the high-fat diet, fasting blood glucose tended to be decreased in mice treated with the alga. *Palmaria mollis* significantly reduced hepatic triglyceride accumulation induced by high-fat feeding, but the control level was not reached. To explain the delipidating effect of the seaweed, the authors measured mRNA levels of several genes involved in hepatic β-oxidation and lipid synthesis. With regard to β-oxidation, peroxisome proliferator-activated receptor alpha (*Pparα*) gene expression was higher in the mice treated with the alga than in the other groups. In the absence of *Palmaria mollis*, *Acox1* gene expression was lower among mice fed with the steatotic diet than in the other groups and Acadm gene expression remained unchanged among the three groups. Regarding lipid synthesis-related genes, whereas sterol regulatory element binding transcription factor 1 (*Srebf1*) gene expression remained unchanged among the three groups, peroxisome proliferator-activated receptor gamma (*Pparɣ*) expression, which was increased by the steatotic feeding, was completely restored by *Palmaria mollis* supplementation. Moreover, CCAAT/enhancer-binding protein alpha (*C/ebpα*) gene expression was higher in the mice treated with the alga than in the other groups. Thus, *Palmaria mollis* showed its ability to suppress hepatic lipid accumulation by the modulation of β-oxidation and of lipid synthesis.

In another study, du Preez et al. [24] analysed the effects of the red algae *Sarconema filiforme* extract that provided 34% of carbohydrates and 12% of proteins in rats. Two groups were fed with either a standard corn starch diet (C) or a high-carbohydrate, high-fat diet (H) for 16 weeks and the other two groups received the same diets but supplemented with 5% of *Sarconema filiforme* extract for the remaining eight weeks. In addition, the drinking water of the rats fed with the high-carbohydrate, high-fat diet was supplemented with 25% fructose. At the end of the experimental period, the high-carbohydrate, high-fat diet significantly increased hepatic fat deposition and infiltration of inflammatory cells, and *Sarconema filiforme* partially prevented these effects. Regarding the serum parameters, glucose levels were higher in the groups that received the high-carbohydrate, high-fat diet than in the groups fed with the control diet. What is more, the algae had no effect on these parameters. In contrast, it did reduce ALT and AST values.

When gut microbiota was analysed, no significant differences were observed in Shannon’s diversity or richness of faecal samples among the experimental groups. However, bacterial community structure was affected by both the diet and the alga. Rats fed with the standard diet had lower *Firmicutes/Bacteroidetes* ratio than those fed with the high-carbohydrate, high-fat diet. Nevertheless, no differences in this ratio were observed in rats supplemented with the alga when compared with their pertinent control group. Compared to the other three groups, the abundance of bacteria from the *Bacili* class and from the *Lactobacillaceae* family was found to be lower in rats fed with the high-carbohydrate, high-fat diet with algae supplementation. One zOTU (zero-radius operational taxonomic unit), which belongs to the *Muribaculaceae* family, was only found in rats not supplemented with *Sarconema filiforme,* whereas another zOTU belonging to the *Ruminococcaceae* and *Desulfovibrionaceae* families was only found in the groups that received the alga. In conclusion, *Sarconema filiforme* supplementation modulated gut microbiota without changing the *Firmicutes/Bacteroidetes* ratio. The correlations between changes in the gut microbiota and physiological changes suggest that this is likely to be one of the mechanisms that explain the beneficial effects of *Sarconema filiforme* on this experimental model.

Lee et al. [25] conducted a study aimed at researching the effects of the red algae *Grateloupia elliptica* using C57BL/6 mice. Animals were distributed into four groups: the control group was fed with a chow diet; the HFD group received a high-fat high-sucrose diet; the L-GEE group was administered the same high-fat high-sucrose diet but supplemented with a low dose of *Grateloupia elliptica* (125 mg/kg body weight/day) and the H-GEE group was fed with the same high-fat high-sucrose diet but supplemented with a high dose of *Grateloupia elliptica* (250 mg/kg body weight/day). The alga extract was administered orally for seven weeks. Both doses of the *Grateloupia elliptica* significantly reduced the hepatic steatosis, although control values were not reached. The authors did not indicate if a dose-response pattern was observed.

Nabil-Adam et al. [26] have recently published a study devoted to assessing the potential of red alga *Gromphadorhina oblongata* on the prevention of LPS-induced liver inflammation and injuries. For this purpose, BALB/C mice were divided into four groups: a negative control group that received a daily intraperitoneal saline solution, an induction control group that was given 5 mg/kg body weight/day of LPS intraperitoneally; a protected group that received 200 mg/kg body weight/day of the seaweed extract for two hours before LPS treatment, and a positive control group that received 200 mg/kg body weight/day of the seaweed extract without LPS. The experimental period length was one week. At the end of this week, serum transaminases (ALT and AST) were higher in the induction control group than in the other experimental groups. Livers displayed an abnormal histopathological appearance as portal lymphoplasmacytic, inflammatory infiltrates, hydropic alterations, portal lymph plasmatic infiltrate and parenchymal hydropic alterations with apoptosis and binucleated cells were found in the induction group. The *Gromphadorhina oblongata* extract protected against LPS effects.

According to the reported studies, it can be concluded that the five red algae analysed were able to prevent liver triglyceride accumulation induced by diets rich in fat and those rich in fat and sugars. This positive effect is found in both rats and mice. The majority of the reported studies did not address the mechanisms of action involved in this effect. Interestingly, two studies that were focused on *Plocamium telfairiae* showed that the positive effects on liver fat accumulation were observed with quite different experimental period lengths (7 and 14 weeks).

#### 2.2.3. Brown Algae

Four brown algae have been researched in the studies reported in the literature: *Undaria pinnatifida, Fucus vesiculosus, Ascophyllum nodosum Sargassum thunbergii* and *Sargassum horneri* (Table 4).

Murata et al. [27] carried out a study using Sprague–Dawley rats fed with a standard diet supplemented or not with a percentage of 0.5, 1, 2, 5 and 10 of *Undaria pinnatifida* for three weeks. There was a reduction in hepatic triglycerides in the groups supplemented with 1, 2, 5 and 10% of the seaweed. In order to understand the mechanism involved in these effects, two metabolic pathways, lipogenesis and β-oxidation, were analysed in the groups supplemented with 5 or 10% *Undaria pinnatifida*. Regarding the lipogenic pathway, Acyl-CoA dehydrogenase (ACAD) and 2,4-dienoyl-CoA reductase (DECR1) activities were increased and glucose-6-phosphate dehydrogenase (G6PD) activity was reduced in both supplemented groups. Concerning lipid oxidation, CPT and ACO were increased only in the group receiving 10% *Undaria pinnatifida*. Based on these results, the authors concluded that *Undaria pinnatifida* supplementation increased the activity of fatty acid oxidation, which is responsible for the lower levels hepatic triglycerides. They also pointed out that it was necessary to examine lipid metabolism in more detail.

A further study was carried out by the same group [28] using the same animal model, but in this case, they administered a high dose of seaweed (19.1%) *Undaria pinnatifida* for a period of four weeks. Liver weight and triglyceride content were lower in supplemented animals than in the controls. With regard to the mechanisms of action, G6PD activity was reduced, whereas ACO and 3-hydroxiacil-CoA dehydrogenase were increased in the *Undaria pinnatifida* group. Contrarily to the effects observed in the previous study, CPT activity remained unchanged, but the authors did not explain this discrepancy.

Using the same brown alga, Li et al. [29] conducted a study using C57BL/6J mice which were distributed in four groups: standard diet group (C), standard diet group with 10% of *Undaria pinnatifida* (NUP), high-fat diet group (HFD) and high-fat diet group with 10% *Undaria pinnatifida* (HUP). After ten weeks of treatment, the glycemic analysis showed higher values in the HFD group than in the C, NUP and HUP groups. The histological analysis showed lager hepatocytes size and fat vacuoles in the HFD group. The analysis of gut microbiota composition revealed that *Undaria pinnatifida* restored the instability in *Firmicutes* and *Bacteroides* induced by the high-fat diet. Related to family level, *Undaria pinnatifida* restored the abundance *of Lachnospiraceae* and *Streptococcaceae.* Furthermore, in HUP there were a higher abundance of *Bacteroidaceae* and lower abundance of *Marinifilaceae*, compared with the HFD group.

Gabbia et al. [30] used a phytocomplex extracted from the brown seaweeds *Fucus vesiculosus* and *Ascophyllum nodosum* to analyse its protective effect on rats fed with a high-fat diet. Wistar rats were randomised into two experimental groups and were administered a high-fat diet (HFD) (60.3% of energy from fat). One of the groups was treated with 7.5 mg/kg body weight of the alga extract by intragastric gavage on a daily basis. Liver weight of animals supplemented with the seaweed extract was significantly lower than that of rats receiving the HFD alone. While HFD rats showed moderate microvesicular steatosis, animals treated with the seaweed extract just showed isolated steatotic hepatocytes. This effect was in accordance with plasma ALT and AST levels, which were significantly decreased. The authors additionally measured postprandial blood glucose levels, and they observed that after a starch-simulated, high-carbohydrate meal, animals receiving the seaweed extract showed a delayed and lower blood glucose peak, indicating that the treatment improved post-prandial glucose control.

Kang et al. [31] studied the effect of the brown algae *Sargassum thunbergii* on lipid accumulation in the liver of mice which displayed steatosis induced by a high-fat diet. C57BL/6 mice were distributed into four experimental groups: the control group was fed with a chow diet, the HFD group received a high-fat diet (45% fat), the ST100 group was administered the same high-fat diet together with 100 mg/kg/d of *Sargassum thunbergii* extract and the ST300 group was fed with the same high-fat diet along with 300 mg/kg body weight/day of *Sargassum thunbergii* extract. The extract was dispensed orally for seven weeks. Hepatic histological analysis revealed that, as expected, the high-fat diet increased lipid accumulation into the liver. When compared to the HFD group, smaller lipid droplets were found in the ST100 and in ST300 groups. This reduction was found in a dose-dependent manner.

Using an alga from the same genus, *Sargassum horneri*, but administered to mice instead of rats, Murakami et al. [32] carried out a study where C57BL/6J mice were fed with a normal diet, a high-fat diet (HF) or a high-fat diet supplemented with either 2% (HF + ShL) or 6% (HF + ShH) of the seaweed. After 13 weeks of treatment, and compared to animals fed with the high-fat diet, both groups receiving the alga showed reductions in body and white adipose tissue weight, as well as in serum glucose and insulin levels. Mice in the HF group developed liver steatosis, as shown by the increase in liver weight and triglyceride content. This effect was also revealed through histological analysis. Supplementation with the seaweed avoided these effects in a dose-dependent manner. Serum ALT and AST levels were also normalised, but only after supplementation with the highest dose. Since animals receiving seaweed supplementation showed higher content of fat in faeces, the authors hypothesised that the reduction in liver triglycerides might have been due to a diminished absorption of dietary lipids.

Among the brown macroalgae, the most frequently analysed is *Undaria pinnatifida,* which has been demonstrated to be effective in both rats and mice after medium length and longer experimental periods. Nevertheless, in the three studies reported, where this type of alga has been used, the diets administered to animals were standard diets providing normal amounts of fat and sugars, the two nutrients that increase liver triglyceride accumulation. Consequently, further studies that make use of the diet that induces steatosis are needed to confirm the preventive effects of this seaweed. In the case of *Sargassum thunbergii,* this seaweed prevents the liver steatosis induced by a high-fat diet, in both mice and rats, in a dose-dependent manner.

## 3. Human Studies

To date, very little information aimed at analysing the effects of algae in humans has been reported. Ebrahimi-Mameghani et al. [33] carried out a double-blind, placebo-controlled, randomised clinical trial in 55 obese patients aged 20–50 years with confirmed NAFLD by ultrasonography. Individuals in the intervention group (29) received 1200 mg/day of *Chlorella vulgaris* dispensed in four tablets of 300 mg and 400 mg/day of vitamin E, whereas the placebo group (26) received 400 mg/day of vitamin E and four placebos for eight weeks. In this study, the only parameters related to the liver were serum transaminases, which did not yield differences between both experimental groups.

Li et al. [34] studied the association of algae consumption with newly diagnosed NAFLD by ultrasound in the adult population. To do so, they carried out a cross-sectional study involving 24,572 adult subjects from The Republic of China. The authors observed that algae consumption, assessed using a food frequency questionnaire, was negatively associated with the prevalence of NAFLD, especially in non-obese patients. Adjustments for several factors were implemented: age, sex, body mass index (BMI), smoking status, alcohol drinking status, socioeconomic status, physical activity, family history of disease (including cardiovascular disease, hypertension, hyperlipidaemia and diabetes), hypertension, hyperlipidaemia, diabetes and total energy intake. Additional adjustments were also applied for “fruits and sweet”, “healthy” and “animal foods” dietary pattern scores. The authors stated that to clarify the causality, more prospective studies and clinical trials were required.

## 4. Concluding Remarks

The reported data described in the present review show that there is scientific evidence supporting the beneficial effects of microalgae and the different types of macroalgae (green, red and brown) on liver steatosis in rodent models (Figure 1). The vast majority of the reported studies analysed the preventive effects, since algae were administered to animals together with the diet that induced liver steatosis. The positive effect has been demonstrated using a wide range of doses in the diet, 7.5 to 300 mg/kg body weight/day or 1 to 10% and experimental periods ranging from 3 to 16 weeks. Different ways of algae administration have been used in the published studies, included in the diet, included in the drinking water or administered by oral gavage, and all of them have been adequate to show the beneficial effects of algae.

Two important limitations of the scientific knowledge available to date are: (a) very few studies have investigated the mechanisms of action underlying the preventive effects of algae on liver steatosis, (b) for the majority of the algae studied, a single paper has been reported and thus, it is not possible to establish the best conditions so as to recognise their beneficial effects and (c) all the studies have been performed in male rodents, and consequently potential sexual dimorphism has not been addressed. In this scenario, further studies are necessary not only to clarify the mechanisms of action of the anti-steatotic effects of algae, but also to analyse their potential effects on the management of liver steatosis when this alteration is already developed.

Moreover, the beneficial effects of algae observed in rodent need to be standardised to humans, before we can start thinking about these products as the new tools in the fight against fatty liver. To date, only two studies have addressed the effects of seaweeds on liver steatosis, a study where *Chlorella vulgaris* was administered as a nutraceutical (tablets) and another study based on the daily algae consumption of the studied population. Despite the great differences in terms of experimental design, both of them revealed positive aspects of algae consumption.

Taking into account that the susceptibility to develop NAFLD depends on genetic background, among other factors [35], it is important to address future studies devoted to analysing potential interactions of algae treatments with genetics and epigenetics, in order to established which subjects can get the most benefit, in the framework of personalised nutrition. Another factor with an important role in the development of NAFLD is gut microbiota [36]. Considering that several components of algae are able to modify microbiota composition [37,38,39], an interesting field of future research is to establish the relationship between these modifications and the improvement of NAFLD produced by algae.

## Figures and Tables

**Figure 1 nutrients-13-02017-f001:**
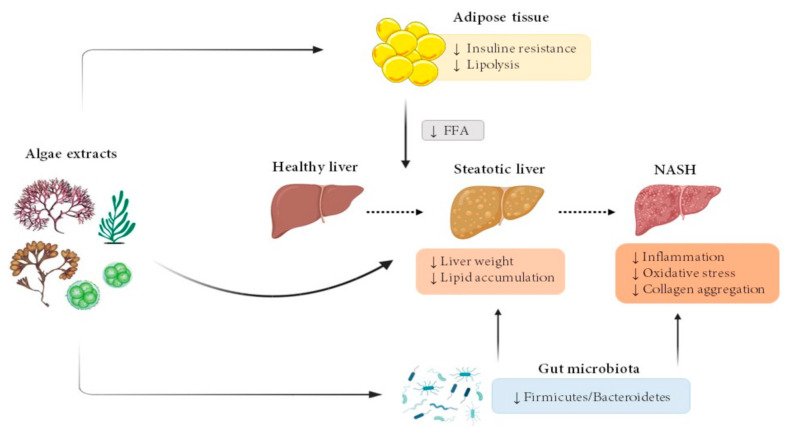
Effects of microalgae and macroalgae extracts on metabolic alterations leading to steatohepatitis. FFA, free fatty acids; NASH, non-alcoholic steatohepatitis.

**Table 1 nutrients-13-02017-t001:** Effects of microalgae extracts in animal models.

Author	Algae Species	Animal Model and Experimental Period Length	Experimental Groups	Effects	Mechanisms
Kumar et al., 2015 [14]	*Scenedesmus dimorphus* + *Schroederiella apiculate* mixture (green microalgae)	Male Wistar rats 8 weeks	Corn starch diet (C) High-carbohydrate high-fat diet (H) Corn starch diet + 5% microalgae mixture (CSC) High-carbohydrate high-fat diet + 5% microalgae mixture (HSC)	↓ Liver weight ↓ Enlargement of fat vacuoles in hepatocytes (HC vs. H) ↓ Inflammation ↓ ALT and AST activities (HC vs. H) Improved glucose tolerance and insulin sensitivity (HC vs. H)	↓ Infiltration of inflammatory cells
Nakashima et al., 2018 [15]	*Euglena gracilis* (green microalgae)	Male STAM mice 27 days	High-fat diet High-fat diet + 3 g/kg BW/day *E. gracilis* High-fat diet + 3 g/kg BW/day Paramylon High-fat diet + 10 mg/kg BW/day Telmisartan	Liver weight: NS Liver TG: NS ↓ Collagen aggregation (*Euglena* vs. vehicle) NAS Score: NS Serum ALT: NS	↓ Immunostaining of F4/80, α-SMA (trend) Inflammation-related genes: NS Fibrosis-related genes: NS
Pham et al., 2019 [16]	*Spirulina platensis* (blue-green microalgae)	Male C57BL/6J mice 20 weeks	Low-fat (LF) High-fat/high-sucrose/high-cholesterol (HF) HF + 2.5% *S. platensis* (HF/SP)	Liver weight: NS Liver TG and cholesterol: NS ↓ Plasma ALT level Improvement of glucose tolerance Hepatic collagen accumulation: NS	mRNA levels of *Col1a1* in liver: NS ↓ *Il-1β* mRNA levels in splenocytes
Mayer et al., 2021 [17]	*Tisochrysis lutea* (brown-golden microalgae)	Male Wistar rats 8 weeks	Standard diet (CTRL) High-fat high-fructose diet (HF) High-fat high-fructose + 12% *T. lutea* (HF-Tiso)	↓ Liver TG and cholesterol ↓ Plasma AST, AST/ALT ↓ Plasma glucose, insulin, leptin ↓ Plasma TNF-α ↓ HOMAR-IR	No information provided

ALT: alanine aminotransferase, AST: aspartate aminotransferase, BW: body weight, Col1a1: collagen type I alpha 1 chain, F4/80: homologue in mouse to epidermal growth factor-like 1 in humans, HOMA-IR, homeostasis model assessment of insulin resistance; Il-1β: interleukin-1β, α-SMA: alpha smooth muscle actin, TG: triglycerides, TNF-α, tumour necrosis factor-α, ↑: increased, ↓: decreased, NS: not significant.

**Table 2 nutrients-13-02017-t002:** Effects of green macroalgae extracts in animal models.

Author	Algae Species	Animal Model and Experimental Period Length	Experimental Groups	Effects	Mechanisms
Sharma et al., 2017 [18]	*Caulerpa lentillifera*	Male C57BL/6J mice 10 weeks	Standard diet High-fat diet High-fat diet + 250 mg/kg BW/day of *C. lentillifera*	↓ Liver weight ↓ Liver TG, TC and FFA ↓ Plasma FFA, glucose and insulin	No information provided
du Preez et al., 2020 [19]	*Caulerpa lentillifera*	Male Wistar rats 16 weeks	Corn starch (C) Standard diet (C) High-carbohydrate high-fat (H) Standard diet + 5% *C. lentillifera* (CCL) High-carbohydrate high-fat + 5% *C. lentillifera* (HCL)	↓ Liver TG content (H vs. HCL) Inflammatory cell infiltration: NS Plasma ALT, AST: NS ↓Firmicutes/Bacteroidetes ratio (H vs. HCL)	No information provided
Song et al., 2018 [20]	*Ulva prolifera*	Male C57BL/6 mice 8 weeks	Standard diet High-fat diet High-fat diet + 2% ethanol extract of *U. prolifera* in drinking water High-fat diet + 5% ethanol extract of *U. prolifera* in drinking water	↓ Liver weight ↓ Liver TG content ↓ Serum insulin ↓ Oxidative stress	↓ *Dgat1* and *Dgat2* liver mRNA levels ↑ *Cpt-1a, Acadm* and *Acox1* liver mRNA levels ↓ Serum IL-1β, IL-6 and TNF-α ↓ *Il-1β, Il-6* and *Tnf-α* liver mRNA levels ↓ Liver ROS ↑ GSH content and GSHPx activity

Acadm: medium-chain acyl-CoA dehydrogenase, Acox1: acyl-CoA oxidase 1, BW: body weight; Cpt-1a: carnitine palmitoyltransferase 1A, Dgat1: diacylglycerol O-acyltransferase 1, Dgat2: diacylglycerol O-acyltransferase 2, FFA: free fatty acids, GSH: reduced glutathione, GSH-Px: glutathione peroxidase, IL-1β: interleukin-1β, IL-6: interleukin-6, ROS: reactive oxygen species, TC: total cholesterol, TG: triglycerides, TNF-α: tumour necrosis factor-α, ↑: increased, ↓: decreased. NS: not significant.

**Table 3 nutrients-13-02017-t003:** Effects of red macroalgae extracts in animal models.

Author	Algae Species	Animal Model and Experimental Period Length	Experimental Groups	Effects	Mechanisms
Kang et al., 2016 [21]	*Plocamium telfairiae*	Male C57BL/6 mice 14 weeks	Standard diet High-fat diet High-fat diet +100 mg/kg BW/day of *P. telfairiae*	↓ Liver steatosis ↓ Serum glucose	No information provided
Lu et al., 2020 [22]	*Plocamium telfairiae*	Male C57BL/6 mice 7 weeks	Standard diet High-fat diet High-fat diet +100 mg/kg BW/day of *P. telfairiae* High-fat diet +165 mg/kg BW/day of *P. telfairiae* High-fat diet + 300 mg/kg BW/day of *P. telfairiae*	↓ Hepatic steatosis (all doses)	No information provided
Nakayama et al., 2018 [23]	*Palmaria mollis*	Male NSY/HOS mice 4 weeks	Standard diet High-fat diet High-fat diet + 2.5% of *P. mollis*	↓ Liver TG content	↑ *Pparα, C/ebpα* and *Acox1* mRNA levels ↓ *Pparɣ* mRNA levels *Acadm* and *Srebf1* mRNA levels: NS
du Preez et al., 2020 [24]	*Sarconema filiforme*	Male Wistar rats 16 weeks	Corn starch diet Corn starch diet + 5% *S. filiforme* High-carbohydrate high-fat diet High-carbohydrate high-fat diet + 5% *S. filiforme* (drinking water of rats fed the steatotic diet was supplemented with 25% fructose)	↓ Liver steatosis and infiltration of inflammatory cells (high-carbohydrate high-fat diet 5% *S. filiforme* vs. high-carbohydrate high-fat diet) Serum glucose: NS ↓ Serum ALT and AST (high-carbohydrate high-fat diet 5% *S. filiforme* vs. high-carbohydrate high-fat diet) Firmicutes/Bacteroidetes: NS	No information provided
Lee et al., 2020 [25]	*Grateloupia elliptica*	Male C57BL/6 mice 7 weeks	Standard diet High-fat high-sucrose diet High-fat high-sucrose diet +125 mg/kg BW/day of *G. elliptica* High-fat high-sucrose diet +250 mg/kg BW/day of *G. elliptica*	↓ Hepatic steatosis	No information provided
Nabil-Adam et al., 2021 [26]	*Gromphadorhina oblongata*	BALB/C mice 1 week	Negative control: saline solution Induction control: 5 mg/kg BW/day of LPS Protected group: 200 mg/kg BW/day of *G. oblongata* 2 h before LPS Positive control: 200 mg/kg BW/day of *G. oblongata* without LPS	↓ Liver injury (inflammation and oxidative stress) ↑ Liver apoptosis ↓ Serum ALT, AST	No information provided

Acadm: acyl-CoA dehydrogenase medium chain, Acox1: peroxisomal acyl-CoA oxidase 1, ALT: alanine transaminase, AST: aspartate transaminase, BW: body weight, C/ebpα: CCAAT/enhancer-binding protein alpha, LPS: lipopolysaccharide, Pparα: peroxisome proligerator-activated receptor alpha, Pparγ: peroxisome proligerator-activated gamma, Srebf1: sterol regulatory element-binding protein 1, TG: triglycerides, ↑: increased, ↓: decreased, NS: not significant.

**Table 4 nutrients-13-02017-t004:** Effects of brown macroalgae extracts in animal models.

Author	Algae Species	Animal Model and Experimental Period Length	Experimental Groups	Effects	Mechanisms
Murata et al., 1999 [27]	*Undaria pinnatifida*	Male Sprague–Dawley rats 3 weeks	Standard diet Standard diet + 0.5% *U. pinnatifida* Standard diet + 1% *U. pinnatifida* Standard diet + 2% *U.pinnatifida* Standard diet + 5% *U. pinnatifida* Standard diet + 10% *U.pinnatifida*	↓ Liver TG content in 1, 2, 5 and 10% groups ↓ Liver TC content in 10% group	↓G6PD activity in 5 and 10% groups ↑ CPT activity in 10% group ↑ ACADs activity in 5 and 10% groups ↑ ACO in 10% group ↑ DECR1 in 5 and 10% groups
Murata et al., 2002 [28]	*Undaria pinnatifida*	Male Sprague–Dawley rats 4 weeks	Standard diet Standard diet + 19.1% *U. pinnatifida*	↓ Liver weight ↓ Hepatic TG, TC and phospholipids levels	↓ G6PD activity ↑ ACO and 3-hydroxiacil-CoA dehydrogenase activities CPT activity: NS
Li et al., 2020 [29]	*Undaria pinnatifida*	Male C57BL/6 mice 10 weeks	Standard diet Standard diet + 10% *U. pinnatifida* High-fat diet High-fat diet + 10% *U. pinnatifida*	↓ Liver steatosis ↓ Glucose levels	No information provided
Gabbia et al., 2020 [30]	*Fucus vesiculosus* + *Ascophyllum nodosum*	Male Wistar rats 5 weeks	High-fat diet (HFD) High-fat diet + 7.5 mg/kg BW/day of *F. vesiculosus* and *A. nodosum*	↓ Liver weight ↓ Microvesicular steatosis ↓ Plasma ALT and AST levels Lower and delayed glucose peak	No information provided
Kang et al., 2020 [31]	*Sargassum thunbergii*	Male C57BL/6 mice 7 weeks	Standard diet High-fat diet High-fat diet + 100 mg/kg BW/day of *S. thunbergii* High-fat diet + 300 mg/kg BW/day of *S. thunbergii*	↓ Lipid steatosis	No information provided
Murakami et al., 2021 [32]	*Sargassum horneri*	Male C57BL/6J mice 13 weeks	Standard diet High-fat diet (HF) High-fat diet + 2% *S. horneri* (HF + ShL) High-fat diet + 6% *S. horneri* (HF + ShH)	↓ Liver weight ↓ Liver TG content ↓ Serum glucose, insulin, ALT, AST, ALP and LAP levels ↑ Serum adiponectin ↓ Serum TNF-α	↓ Pancreatic lipase activity

ACADs: acyl-CoA dehydrogenases, ACO: acyl-CoA oxidase, ALT: alanine transaminase, ALP: alkaline phosphatase, AST: aspartate transaminase, CPT: carnitine palmitoyltransferase; DECR1: 2,4 dienoyl-CoA reductase, G6PD: glucose-6-phosphate dehydrogenase, LAP: leucine aminopeptidase, TC: total cholesterol TG: triglycerides, TNF-α: tumour necrosis factor-α, ↑: increased, ↓: decreased, NS: not significant.

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
