# Peer review of "Effect of Microalgae and Macroalgae Extracts on Non-Alcoholic Fatty Liver Disease"

_nutrients, 2021, doi:10.3390/nu13062017_

Round 1
Reviewer 1 Report
Manuscript ID: nutrients-1243758
Dear Editor and Authors,
The manuscript entitled “Effect of microalgae and macroalgae on non-alcoholic fatty liver disease” is a short review on a very interesting topic. The manuscript has few typos, but I have some issues that need to be addressed before its publication.
The authors need to state clearly the objectives of this review: if they are excluding bioactive compounds, only to provide data on extracts they also need to say they are excluding in vitro experiments.
If the authors only review the works concerning extracts, this needs to be highlighted right in the title of the manuscript and in the title of each table. I understand the relevance of giving specific attention to the extracts, but I think it would be interesting that authors justify their choice of excluding the compounds that could be responsible for the bioactivities observed.
The introduction is very short and needs to provide more detail, especially regarding the biotechnological potential of the chosen matrices (algae).
Lines 51-52: Re-write this sentence, as the studies are not “reported”, but instead the studies are “published”.
Figures 1 and 2 are basic. Please, improve them or just explain the information on the main text. Besides, the information is repetitive from what is presented on Tables.
Figure 2 caption is not correct.
In Tables, instead of giving emphasis only to the animal model used, I would suggest the authors to create a column for the algae species, as it is the focus of the review, not the animal models themselves.
The main text needs to be improved. It is merely the results obtained by the original research work without providing a further analysis by the authors. Sometimes it is too much detailed and no connection between the targets that the works explore and the NAFL disease is effectively explored.
Another concern is that the authors write this review as they are talking only to experts, not giving general information for a different target audience.
Author Response
Reviewer 1
The manuscript entitled “Effect of microalgae and macroalgae on non-alcoholic fatty liver disease” is a short review on a very interesting topic. The manuscript has few typos, but I have some issues that need to be addressed before its publication.
The authors need to state clearly the objectives of this review: if they are excluding bioactive compounds, only to provide data on extracts they also need to say they are excluding in vitro experiments.
In vitro experiments in hepatocytes treated with algae extracts have not been published. According to the reviewer’s comment, this information has been provided in the Introduction section, as well as the fact that both in vitro and in vivo studies performed with individual components of algae extracts, are not included in the review (page 2, lines 68-71).
If the authors only review the works concerning extracts, this needs to be highlighted right in the title of the manuscript and in the title of each table. I understand the relevance of giving specific attention to the extracts, but I think it would be interesting that authors justify their choice of excluding the compounds that could be responsible for the bioactivities observed.
The title of the manuscript, as well as the titles of the tables, has been changed in this revised version in order to highlight that the review is based in extracts.
In order to know which compounds are responsible for the effect of algae extracts it is necessary to determine the exact extract composition and to use the corresponding amount of each component in other in vivo studies, using exactly the same experimental conditions. In addition, different combinations of the individual ingredients should be tested in order to assess potential additive or synergistic effects. These studies have not been published in the literature. Indeed, that we can find are studies that analyse the effect of individual components without any reference to the algae extracts that contain these molecules. Consequently, conclusions concerning the compounds that are responsible for the algae extracts cannot be drawn.
We provide this explanation to the reviewer, but we consider not appropriate to include it in the manuscript text.
The introduction is very short and needs to provide more detail, especially regarding the biotechnological potential of the chosen matrices (algae).
The introduction has been extended in this revised version (page 1, lines 28-33; page 1, lines 38-40; page 2, lines 49-65; page 2, lines 68-71). According to the reviewer's comment, the biotechnological potential of algae has been included.
Lines 51-52: Re-write this sentence, as the studies are not “reported”, but instead the studies are “published”.
The suggestion of the referee has been taken into account in this revised version of the manuscript (page 2, line 83).
Figures 1 and 2 are basic. Please, improve them or just explain the information on the main text. Besides, the information is repetitive from what is presented on Tables.
The reviewer is right. We have eliminated these two Figures; in fact the information that they provide is also provided in the text. Instead, we have added a new Figure, showing the metabolic alterations leading to liver steatohepatitis (page 2, lines 75).
Figure 2 caption is not correct.
The referee is right. Nevertheless, as indicated in the previous comment, this Figure 2 has been deleted, according to the reviewer's comment.
In Tables, instead of giving emphasis only to the animal model used, I would suggest the authors to create a column for the algae species, as it is the focus of the review, not the animal models themselves.
Following the reviewer's suggestion, to give more emphasis to the algae species, a new column has been created specifically for the algae species and type.
The main text needs to be improved. It is merely the results obtained by the original research work without providing a further analysis by the authors. Sometimes it is too much detailed and no connection between the targets that the works explore and the NAFL disease is effectively explored.
In the first version, at the end of each alga type section, a paragraph summing-up the main data published concerning this alga type and including several authors' appreciations was already included:
Microalgae (page 5, lines 163-175 in the revised version)
To sum up, after reviewing the literature concerning the effects of microalgae on NAFLD, it can be pointed out that, although the results reported are encouraging, the number of studies is still very scarce. In addition, the reported studies address the effects of different microalgae, and thus only one source of information is available for each one. According to the experimental design used, all the reported studies analysed the preventive effects of microalgae because these were included in the diet that induced liver alterations. Although in all of them microalgae improved several alterations, there is a lack of consensus on the specific effects observed. Thus, whereas the three studies that analysed this parameter reported a reduction of liver inflammation, only two of these studies described a reduction of liver triglyceride content (the other two studies did not find significant changes). Moreover, two of the studies analysed the effects of fibrosis and only one observed a significant improvement. In this scenario, further research is needed in order to assess the effects of microalgae on NAFLD.
Green algae (pages 7-8, lines 236-244 in the revised version)
In summary, as in the case of microalgae, the number of studies addressing the preventive effect of green algae on NAFLD is still scarce. The three reported works (two of them carried out with Caulerpa lentillifera) revealed a reduction in hepatic triglyceride accumulation when animals were fed with a high fat diet, but only one of the studies addressed several aspects of the potential mechanisms of action (insulin resistance, fatty acid and triglyceride metabolism, oxidative stress). In one of these pieces of research, the authors analysed the effects of seaweed supplementation on gut microbiota composition, but they did not establish a clear relationship between these changes and the effects on liver steatosis, but just a significant correlation.
Red algae (page 12, lines 342-348 in the revised version)
According to the reported studies, it can be concluded that the five red algae analysed were able to prevent liver triglyceride accumulation induced by diets rich in fat and those rich in fat and sugars. This positive effect is found in both rats and mice. The majority of the reported studies did not address the mechanisms of action involved in this effect. Interestingly, two studies that were focused on Plocamium telfairiae showed that the positive effects on liver fat accumulation were observed with quite different experimental period lengths (7 and 14 weeks).
Brown algae (page 16, lines 424-432 in the revised version)
Among the brown macroalgae, the most frequently analysed is Undaria pinnatifida, which has been demonstrated to be effective in both rats and mice after medium length and longer experimental periods. Nevertheless, in the three studies reported, where this type of alga has been used, the diets administered to animals were standard diets providing normal amounts of fat and sugars, the two nutrients that increase liver triglyceride accumulation. Consequently, further studies that make use of the diet that induces steatosis are needed to confirm the preventive effects of this seaweed. In the case of Sargassum thunbergii, this seaweed prevent the liver steatosis induced by a high-fat diet, in both mice and rats, in a dose-dependent manner.
The reviewer says that sometimes the text is too much detailed, and thus we have deleted the information concerning animal age and sex for each study description. We have also eliminated data about serum lipid parameters and, in some studies, experimental groups treated with products different from algae. Lastly, in this revised version we have not provided information concerning some methodologies such as liver staining.
We have also reorganised the text in order to present together all the studies that have used the same alga.
Another concern is that the authors write this review as they are talking only to experts, not giving general information for a different target audience.
We believe that the vast majority of the scientists interested in this review will be working on the metabolic effects of algae, and thus they will not have any problem to understand the text. For those with an expertise no so close to this topic, we have tried to add some explanations. For instance, when we present the study published by Pharm et al, and we say that the authors observed that Spirulina platensis improved slightly glucose tolerance, we added the following sentence "which is a positive effect taking into account that a bad glycaemic control is a risk factor for liver steatosis development"

Reviewer 2 Report
I have only few minor points that the Authors may willing to address:
1) The introduction may benefit also of focussing on important contributions in the field of NAFLD i.e. DOI: 10.1016/j.jhep.2018.10.033, of discussing the new nomenclature 'MAFLD' (i.e. https://doi.org/10.1002/hep.31420), and of comments regarding the possible effects of algae only in MAFLD patients and not in NAFLD ones.
2) In the main body of the Review, the Authors might compare the differences between the introduction of algae into the diet (as dietary supplementation) and the consumption of pills containing them.
3) The Authors might comment breifly the differences in dietary habits in the last years, for instance, concerning the increased consumption of sushi in the western countries.
4) What about the genetics? Might the Authors hypothesize a specific effect of the algae-enriched diet based on the genetic background? See also doi: 10.3390/ijms21082986. and possible interactions with gut microbiota? (see also doi: 10.3390/nu11112642).
Author Response
Reviewer 2
I have only few minor points that the Authors may willing to address:
1) The introduction may benefit also of focussing on important contributions in the field of NAFLD i.e. DOI: 10.1016/j.jhep.2018.10.033, of discussing the new nomenclature 'MAFLD' (i.e. https://doi.org/10.1002/hep.31420), and of comments regarding the possible effects of algae only in MAFLD patients and not in NAFLD ones.
The reviewer's suggestion, as well as the references suggested, has been included in this revised version of the manuscript (page 1, lines 28-33; page 1, lines 38-40).
2) In the main body of the Review, the Authors might compare the differences between the introduction of algae into the diet (as dietary supplementation) and the consumption of pills containing them.
Following the reviewer's suggestion, this issue has been included in the Concluding remark section (page 17, lines 461-464; page 17, lines 476-480).
3) The Authors might comment briefly the differences in dietary habits in the last years, for instance, concerning the increased consumption of sushi in the western countries.
New information concerning this issue has been added in this revised version (page 2, lines 49-65).
4) What about the genetics? Might the Authors hypothesize a specific effect of the algae-enriched diet based on the genetic background? See also doi: 10.3390/ijms21082986. and possible interactions with gut microbiota? (see also doi: 10.3390/nu11112642).
There are no data published about the potential effects of algae depending on genetic or epigenetic backgrounds. Nevertheless, according to the reviewer's comment, we have mentioned both issues, genetic background and microbiota, as well as both references, at the end of the Concluding remarks (page 17, lines 481-488).

Round 2
Reviewer 1 Report
Manuscript ID: nutrients-1243758-R1
Dear Editor and Authors,
First, I would like to acknowledge the improvements made by the authors to its original submission. There are still some minor changes that need to be done.
Line 59: Replace “y” by “and”
Lines 66-71: Although I acknowledge the literature review made by the authors, I would suggest them to replace the sentence “No in vitro studies carried out with algae extracts in cultured hepatocytes have been found in the literature to date.” By “To the best of our knowledge, no in vitro studies with algae extracts in cultured hepatocytes have been carried out to date”. Plus, I would prefer to see the objective of the review at the end of the paragraph. The authors provide some rationale for choosing only algae extracts so I would remove the sentence “On the other hand, both in vitro and in vivo studies performed with individual components of algae extracts, are not included.”
In sum, and if the authors intend so, I suggest this paragraph to be like this:
“To the best of our knowledge, no in vitro studies with algae extracts in cultured hepatocytes have been carried out to date. The present review aims to gather scientific evidence regarding the beneficial effect of microalgae and macroalgae extracts on hepatic steatosis in preclinical and clinical studies, as well as the potential mechanisms involved in these effects.”
Line 74: Replace “summaries” by “summarizes”
Figure 1 was much improved; however, the abbreviation of NASH is not included in the figure caption.
Author Response
Line 59: Replace “y” by “and”
It has been replaced.
Lines 66-71: Although I acknowledge the literature review made by the authors, I would suggest them to replace the sentence “No in vitro studies carried out with algae extracts in cultured hepatocytes have been found in the literature to date.” By “To the best of our knowledge, no in vitro studies with algae extracts in cultured hepatocytes have been carried out to date”. Plus, I would prefer to see the objective of the review at the end of the paragraph. The authors provide some rationale for choosing only algae extracts so I would remove the sentence “On the other hand, both in vitro and in vivo studies performed with individual components of algae extracts, are not included.”
In sum, and if the authors intend so, I suggest this paragraph to be like this:
“To the best of our knowledge, no in vitro studies with algae extracts in cultured hepatocytes have been carried out to date. The present review aims to gather scientific evidence regarding the beneficial effect of microalgae and macroalgae extracts on hepatic steatosis in preclinical and clinical studies, as well as the potential mechanisms involved in these effects.”
As suggested by the reviewer, the paragraph has been modified.
Line 74: Replace “summaries” by “summarizes”
It has been replaced.
Figure 1 was much improved; however, the abbreviation of NASH is not included in the figure caption.
The abbreviation of NASH has been included.
